# A Low-Cost Luxometer Benchmark for Solar Illuminance Measurement System Based on the Internet of Things

**DOI:** 10.3390/s22197107

**Published:** 2022-09-20

**Authors:** Omar Guillán Lorenzo, Andrés Suárez-García, David González Peña, Manuel García Fuente, Diego Granados-López

**Affiliations:** 1Research Group Solar and Wind Feasibility Technologies (SWIFT), Electromechanical Engineering Department, University of Burgos, Avda. Cantabria s/n, 09006 Burgos, Spain; 2Defense University Center, Spanish Naval Academy, University of Vigo, Plaza España s/n, 36920 Marín, Spain

**Keywords:** Internet of Things (IoT), luxometer, natural illumination, low-cost sensor

## Abstract

Natural illumination has an important place in home automation applications. Among other advantages, it contributes to better visual health, energy savings, and lower CO_2_ emissions. Therefore, it is important to measure illuminance in the most accurate and cost-effective way. This work compares several low-cost commercial sensors (VEML 7700, TSL2591, and OPT3001) with a professional one (ML-020S-O), all of them installed outdoors. In addition, a platform based on the Internet of Things technology was designed and deployed as a centralized point of data collection and processing. Summer months have been chosen for the comparison. This is the most adverse situation for low-cost sensors since they are designed for indoor use, and their operating range is lower than the maximum reached by sunlight. The solar illuminance was recorded every minute. As expected, the obtained bias depends on the solar height. This can reach 60% in the worst circumstances, although most of the time, its value stays below 40%. The positive side lies in the good precision of the recordings. This systematic deviation makes it susceptible to mathematical correction. Therefore, the incorporation of more sensors and data that can help the global improvement of the precision and accuracy of this low-cost system is left as a future line of improvement.

## 1. Introduction

Energy efficiency is one of the biggest concerns nowadays. During the last months, a drastic increase in electrical energy prices has been seen. The main causes are the strong revaluation of natural gas on international markets and the rise of CO_2_ emission market prices. Therefore, reducing dependence on fossil fuels is a matter of urgency. The reduction in electricity consumption would help in following this direction. The use of natural illumination in buildings could play a vital role in decreasing energy dependency; hence, it is essential to measure solar illuminance.

For the analysis of energy efficiency in buildings, knowing the values of solar irradiance and illumination is of crucial importance. The estimation of the light reaching a given work area inside a building requires knowledge of the outdoor lighting conditions. The International Commission on Illumination (Commission Internationale de l’Éclairage, CIE) proposed a characterization of skies taking into account the illuminance along the celestial dome. In it, three classes of skies (cloudy, partly cloudy, and clear) are contemplated, each subdivided into five types. Consequently, it proposes a total of 15 models of the celestial dome for different conditions of cloudiness and, therefore, illumination [1]. At present, the most reliable way to analyze the type of sky, according to the CIE, is to use an expensive device that divides the sky dome into 145 sectors and measures the luminous emittance of each of them sequentially; such a device is called a “sky-scanner” [2]. Essentially, it is a pyranometer that is mechanically orientated into prefixed azimuths and altitudes. This process, in addition to lasting several minutes, puts much wear on the device. Its acquisition and maintenance costs make it difficult to make it widely available. This work is the first step in replacing the sky-scanner with a set of preoriented low-cost illuminance sensors. Being able to find out which sensors perform better will help in the design and implementation of a similar device at a significantly lower price. For that purpose, we compare several inexpensive illuminance sensors against a high-cost professional luxometer. The results show good precision and regular accuracy.

Sensor networks are consolidating as promising data sources for future forecasting applications, particularly for local and very short-term applications. This trend has been reinforced by the developments in big data applications based on information sharing among multiple systems [3]. These networks will be subject to environmental degradation, dropped packets, and changes in topology. For that reason, the sensing network must efficiently send traffic, be robust against network device failures, and be resistant to the environment. In other words, the network should be made of inexpensive, simple, distributed devices that cooperate to deliver data over a large area for an extended period of time. Current solutions exist for self-sustaining wirelessly networked modules [4,5] but do not attempt to achieve high spatial density over the harsh environment of building rooftops. Consequently, our goal is not only to find whether it is possible to obtain good illuminance measurements with cheap off-the-shelf sensors but also to automate and integrate the measurement process into the so-called Internet of Things (IoT) ecosystem—send data from different locations to a centralized point, where they will be stored into a Time Series Database (TSDB), analyzed, processed, and finally displayed in ubiquitous web dashboards. The number of IoT networks has increased dramatically in the last decade. This is mainly due to two factors: the low cost of the sensors required for massive data collection and the affordability of the computing infrastructure capable of handling the volume of data generated in an agile manner. Regarding IoT, in relation to the subject matter at hand, there has been an increase in the number of publications in recent years on indoor lighting [6,7] and multispectral analysis [8,9,10]. This work follows this new trend that will surely continue for years to come.

We propose a novel integrated method to obtain spatially distributed illuminance measurements from inexpensive, widely available sensors and compare their accuracy and precision against a lab-grade luxmeter. The paper will be structured as follows: A complete description of the selected professional sensor, low-cost sensors, electronic circuit design, and the deployed IoT infrastructure will be presented in Section 2. In Section 3, the measurements of the low-cost and professional sensors will be compared. Finally, the conclusions will be presented in Section 4.

## 2. Methodology

The present study has been divided into three phases (Figure 1): the development of a networked wireless data acquisition device integrating the low-cost sensors, the acquisition of data during the summer months of 2022, and the comparison of the low-cost sensor measurements with the professional luxmeter. The development has taken place over a short time span, prioritizing agile development and rapid delivery of results. This is the first stage of a larger project. The ultimate goal will be to obtain a device for the characterization of skies according to the nomenclature of the CIE taxonomy. An attempt has been made to answer quickly whether existing sensors can integrate such a platform or whether a step back is required for further development. In the case of poor results, it would be recommended to abandon this path and look for other ways to accurately measure solar illuminance at a low cost.

In this section, a series of key aspects for the development of the present study are discussed. First, the main characteristics of the professional luxmeter used are presented. It is the reference. Its measurements should be replicated by the low-cost luxmeters as accurately as possible. Second, the selected low-cost luxmeters will be discussed in detail. Their characteristics and limitations will be presented. Third, the prototype setup to record the measurements of the low-cost sensors is detailed. Finally, the IoT platform and the software developed for data collection are shown.

### 2.1. Profesional Luxmeter

The reference sensor used is the EKO ML-020S-0. In Table 1, its most relevant characteristics for the present comparison can be found. It is observed that it produces a voltage signal of tens of millivolts. This makes it necessary to have special electronics capable of capturing the oscillations of such low voltage values. This signal conditioning device incurs an additional cost to the amount required for installation. It will be seen later that low-cost sensors are designed to operate in wider voltage ranges. This results in a lower overall cost. In addition, an error estimate for a temperature range of −10 to 50 °C is highlighted. Higher temperatures are to be expected in a device installed outdoors without any cooling. Nevertheless, its low value gives an idea of the accuracy of this device.

### 2.2. Low-Cost Sensors

A set of three light sensor modules was chosen. Table 2 summarizes the main characteristics of each sensor. A detailed search of illumination sensors in the market was carried out. Those with the widest possible illuminance range were selected. The ones finally chosen were the VEML7700 [11], TSL2591 [12], and OPT3001 [13]. All of them have been previously used in IoT projects [14,15]. Only the VEML7700 achieves the maximum solar luminance estimated at 120 klx. It is also noted that some models are capable of measuring the solar infrared (IR) spectrum in addition to visible light. However, these measurements were not used in this study.

In the spectral response provided by the manufacturers of the low-cost sensors (Figure 2), it can be seen that the VEML7700 sensor (light blue color) is the one that best represents the responsivity of the human eye (red color) to the visible light spectrum. In addition, this sensor has a very smooth behavior. This contrasts with the responsivity of the OPT3001 sensor, which, although it has a similar wavelength range to the VEML7700, has a curlier transfer function. The discordant note is given by the data found in the TSL2591 data sheet (yellow color). This device has two channels and measures both visible and infrared light. The libraries provided by the manufacturer are in charge of doing the transformation to illuminance. The exact combination of the two channels used is unknown. Therefore, it has not been possible to obtain a single response function as in the case of the OPT3001 and VEML7700, making an a priori comparison with the responsivity of the human eye unfeasible.

For comparison purposes, the curves were digitized, and the comparison was made directly on the values obtained. The manufacturers only provide such data in the form of graphs in their data sheets. Therefore, there may be some minor errors caused by digitization. These errors should not discourage the use of a low-cost sensor to replace a professional one. It should be mentioned that the unit used is the lux, and it corresponds to the luminous flux weighted according to the response of the human eye. This is related to the area under the response curve. Therefore, an overall magnitude must be compared, not the individual response to each wavelength. The average relative error of the OPT3001 is −8.8%, while the VEML7700 is 12.1%. A similar magnitude error should be observed in the illuminance measurements of both sensors. Observing the relative error in the response of both sensors with respect to the human eye (Figure 3), the OPT3001 sensor performs better at the analyzed wavelengths. It behaves in a more stable way. The VEML7700 is penalized by the displacement of the maximum peak of its response curve, explaining its sinusoidal shape. The 100% error values reached at low wavelengths are due to the fact that, although the response of the human eye is very low, the response of the sensors is zero. It has not been possible to add the values of the TSL2591 to the comparison because the manufacturer did not provide the necessary data, as previously mentioned.

### 2.3. Prototype

A PCB board (Figure 4) was designed so light intensity values can be obtained simultaneously from the three different low-cost sensors and finally compared against a professional, calibrated luxmeter (EKO ML-020S-O). The board is designed around Espressif’s ESP8266 microcontroller because of its Wi-Fi communication capabilities, well-documented API, and low cost [16]. The three lux sensors are connected to the microcontroller via a well-known I2C protocol. None of these sensors share the same I2C address; nevertheless, an I2C multiplexer (TCA9548A) was included in the system for the sake of future prototype expandability. The PCB is completed with a real-time clock (DS3231) and a micro-SD slot: in the event of network failure, or in the case the device is used as a nonconnected standalone datalogger, the measurements, along with their timestamp, are still recorded in JSON format inside the micro-SD card [17]. Every sixty seconds, the microcontroller sequentially polls all three sensors; the illuminance values, along with the timestamp and other values of interest (integration time, gain, etc.), are included in a JSON structure that is both written to a file and sent via Wi-Fi.

The ESP8266 microcontroller was an Adafruit Feather HUZZAH USB-enabled development board. It is a device widely used for sensing purposes in IoT developments [18,19]. This 32-bit RISC CPU clocked at 80 MHz, with 160 kB of RAM, and 4 MB of external flash memory, in addition to the customary general purpose I/O pins and SPI, UART, and I2C peripherals, includes full IEEE 802.11 b/g/n Wi-Fi compliance thanks to its integrated TR switch, low noise amplifier, power amplifier, impedance adaptation network, and antenna. Since the selected microcontroller does not provide an internal real-time clock (RTC), we included an external one, so in the event of a network failure that prevents network time synchronization, a local timestamp can still be generated. For this task, we chose a module that integrates Maxim DS3231 high-accuracy RTC (thanks to a temperature-compensated internal oscillator) and Atmel AT24C32 4 kB EEPROM, both sharing the I2C bus and backed up with a CR2032 lithium battery. To be fully resilient to a network outage, sensor data must also be stored locally, so a 32 GB micro-SD card is included. This card is inserted into a holder module that talks to the microcontroller by means of the Serial Peripheral Interface (SPI) protocol, which allows for synchronous serial two-way communications. An I2C multiplexer (Texas Instruments TCA9548A) is used in order to address sensors with the same I2C address, which would otherwise overlap. Lastly, the prototype was powered using a Traco TSR 1-2450 step-down switching regulator, capable of providing a steady output of 5 V at 1 A with an input voltage ranging between 6.5 V and 36 V. It has a 94% efficiency, so it is suitable to plug into a 12 V DC power supply or a 12 V NiMH battery, should it be needed.

### 2.4. Internet of Things

The diagram in Figure 5 shows all the IT infrastructure involved in managing the data produced by the sensor board. Using the TCP/IP 7-layer protocol MQTT (Message Queue Telemetry Transport) [20], data are sent to a centralized message broker (Mosquitto) [21]. This method enables the seamless addition of extra sensor boards across several locations. Node-Red acts as a data bridge between the message broker and the database [22]. It also allows the preprocessing of the received data, should that be required. Data are then stored in InfluxDB—a Time Series Database (TSDB)—which is especially suitable for holding and performing real-time analysis of big volumes of time series [23]. Finally, there is Grafana, a web environment capable of displaying rich visual dashboards and graphs based on queries on the TSDB [24]. The data displayed on such graphs can be easily exported into CSV to perform further mathematical modeling and analysis.

Since the tested sensors are heterogeneous, for each one, a variable of type struct is defined, whose members are the different parameters provided by the sensor under test (e.g., lux, human-visible ambient light, IR, etc.). In order for the board to be able to work offline, besides sending the measured values through Wi-Fi (and getting the current timestamp via Network Time Protocol -NTP-), the embedded micro-SD card slot and real-time clock can be used as a network-isolated datalogger, which will save one text file per day with the same JSON-formatted information as it would be sent via wireless network, should it be available. The program developed for the ESP8266 microcontroller in C++ language using Arduino API is briefly described in the flowchart from Figure 6. In the initialization section (which is only executed after microcontroller boot-up), several functional groups are configured or given operational values: microcontroller’s serial and I2C internal peripherals, light sensors, and external RTC. The latter is configured with the program’s compilation date and time, which will be held thanks to the customary CR2032 coin cell battery. A timer interruption is set to trigger every 60 s. In its interruption service routine (ISR), a global boolean variable is toggled so that when the program waits for the next measurement cycle, it does not have to do an active wait, thus not letting Wi-Fi routines starve, keeping the connection up. The microcontroller’s Wi-Fi internal card (RF PHY + MAC) is configured in station mode (ST), so it can connect with the access point (AP) generated by the server. Once network layers 1 to 4 from the OSI model are established between the microcontroller and the server, a message queue telemetry transport ( MQTT) client is enabled in the former. This layer 5–7 protocol will be responsible for sending sensor data to the server (which will act as an MQTT broker) in an efficient way (in terms of computational resources, bandwidth, and energy consumption). In the measurement stage, every sensor is polled, and its information is temporarily saved to its data struct variable. Once the polling has finished, the next stage begins, in which all data from every sensor are structured into a JSON object. This lets us both send it via an MQTT Publish message to the central broker and also save it in the on-board micro-SD card (one file per day, one JSON-formatted line per multiple-sensor measurement). Finally, the program reaches a stage in which it is waiting for either the timer interruption (which will trigger a new measurement) or any interruption from the network stack.

All the described applications (Mosquitto, Node-Red, InfluxDB, and Grafana) are deployed inside a Linux server as Docker containers [25]. Docker is a tool that packages an application and its dependencies into a virtual container that can be executed on any Linux server. We decided to leverage such containers because of their portability, flexibility, and ease of deployment without the hassle or resource waste that would involve a similar approach based on virtual machines. As can be seen in Figure 7, for three of these containers, its network interface (eth0) is configured with its own private address in the 172.17.0.0/24 range. A software bridge (“docker0”) is enabled so containers connected to it can communicate among themselves while the remaining other remains isolated. This bridge is finally attached to the Linux routing daemon and routed through the server’s Ethernet port (eth0). This interface is set up with an IP address belonging to the University of Burgos (UBU) private network (10.168.168.46), and it can be accessed through public internet via a provided VPN. Because of the security policies, the sensor board cannot be wirelessly connected to the UBU network, and a workaround had to be implemented: a Wi-Fi hotspot (“Luxometers”) is configured using the server’s second network interface (wlan0); the sensor board connects to it and receives private addressing in the range 10.42.0.0/24 from server’s internal DHCP. The physical interface is directly assigned to the Mosquitto container, which, as stated previously, is the MQTT broker and, thus, the unique point that directly talks to the sensor board. Both physical interfaces of the server (wlan0 and eth0) can communicate among them through the server’s routing daemon.

### 2.5. Data Campaign

The experimental setup used in this work has been installed on the roof of the Higher Polytechnic School at Burgos University (42°21′04″ N; 3°41′20″ O; 856 m above mean sea level). Figure 8 shows the sky scanner equipment and its geographical location. Burgos has an average of 575 mm of precipitation and a global irradiance of 1500 kWh/m^2^ per year, as can be seen on the typical meteorological year (TMY) defined by the Spanish State Meteorology Agency (AEMET) in the last twenty years [25]. The analyzed period of time is the months of June 2022 and July 2022. At first, this could be criticized because of the short amount of data gathered. In meteorological analyses, it is customary to make use of an entire year. However, the physical magnitude herein measured depends on the solar altitude. The sun reaches its highest altitude during the summer solstice on the 21st of June. During this month, the range of the solar illuminance measured is the widest possible. Therefore, the contrast between the professional sensor and the low-cost sensors is the strongest. Having measures of other months would give us differences in temperature or relative humidity. The analysis of these factors will be the subject of future works when more data are available.

## 3. Results

Figure 9 shows the measurements recorded by all sensors on 26 June 2022. It was a sunny day, and the sun almost reached its annual maximum altitude, thus making it a good day for showing the behavior of the sensors through all the solar illuminance ranges. The limitation of the illuminance range in the recorded measurements is clearly visible. While the professional luxmeter reaches 100 klx without any problem, the low-cost sensors hardly reach that figure. They suffer attenuation as the sun increases its altitude and, consequently, the illuminance value increases. The OPT3001 sensor experiences the most anomalous behavior since it stops the normal operation at 75 klx. After that threshold, the sensor stops logging measures or writes down some erratic values. It could be due to an error in the assembly or operating mode that can be modified. However, it is observed how the TSL2591 does not interrupt the measurement process despite having a measurement range very similar to the OPT3001. Although, when the illuminance range of the TSL2591 is exceeded, its measurements are limited, and this maximum value is not exceeded. In the OPT3001, the manufacturer may have chosen not to produce any output because of the lack of accuracy. Finally, also note the inappropriate performance of the VEML7700. It supposedly has a range capable of mimicking the EKO. However, it is clearly seen that the records are below the EKO at times of the highest solar altitudes. In its favor is the fact that at no time does it stop measuring or constraining its records. The VEML7700 is the sensor that most faithfully reflects the evolution of solar illuminance values. Because of these anomalies, the authors will use several units of the same sensor in future works. This will help to perform further analysis on how each of these sensors behaves when the existing illuminance is out of range.

Figure 10 shows the comparison of the low-cost sensor measurements against the calibrated one. Each subplot contains more than 4500 samples. The samples are represented as points with a certain transparency. It is a visual and intuitive way to show where the bulk of the measurements are. If it had not been used, all the points would appear with the same weight, thus giving the false impression of greater dispersion of measurements than the one that actually exists. The black dashed line is the identity with the measurements of the EKO sensor represented in the x-axis. The best performance is undoubtedly shown by the OPT3001, which is quite faithful to the line representing the identity of the EKO professional luxmeter. This good fidelity is obscured by its short operating range. It has been shown that the sun can reach more than 120 klx, and a similar range would be necessary. However, it can be clearly seen that the TSL2591 and the VML7700 measurements are below the professional sensor. An interesting alternative could be to reduce this range by putting the sensor behind a photographic filter. Hence, the behavior of all the sensors in the 0 to 80 klx range would be very regular. It could be modeled using a mathematical approach. All of them have a very consistent precision.

The good accuracy of these sensors makes them susceptible to being used for professional sensor replacement. Their almost systematic bias could be dealt with using a mathematical treatment according to the wavelength. In this way, functionality similar to that of the professional sensor could be obtained. This would allow a larger deployment at two orders of magnitude lower cost. One of these IoT sensors costs less than ten euros, while the professional one we selected costs several hundred. Obviously, if the data were used for the development of high-precision physical models, the shown error could disqualify the low-cost ones. In such a case, depending on the maximum permissible error, even a better sensor than the professional one presented here might be needed. Given the results presented here, there are good indications that low-cost illuminance sensors can achieve performance close to a professional sensor.

Analyzing the relative errors (Figure 11), it can again be seen that the sensors measure below the reference value. At first glance, it could be said that the OPT3001 has the best performance as the mode of the relative errors is the smallest. However, it has a higher dispersion, with some values close to 100%. This behavior is related to its operating range, as will be seen later. It is also found that its error distributions are not normal. Describing the found behavior in a simple way, the relative error for low illuminance values is different from the one for high values. Illuminance is associated with solar height. Therefore, as was shown in Figure 9, the relative errors of the sensors depend on the solar altitude.

Figure 12 shows the low-cost sensors’ relative errors grouped by the solar altitude. The colored area represents the 95% confidence interval of the relative error for each solar altitude interval. The software employed for the graphical representation uses bootstrapping to compute confidence intervals [26]. Bootstrapping estimates the properties of an estimate, such as its variance, by constructing a number of resamples with a replacement of the observed data set [27]. Their small size again demonstrates the high precision of the measurements recorded by these sensors. In addition, a certain bias is again reflected, which could be corrected by taking into account the solar altitude. Another interesting aspect is the good performance of the OPT3001 at low solar altitudes and its deterioration as the solar altitude increases. This behavior is the opposite of the TSL2591 and VML7700 sensors, which perform better at high solar altitudes. This way of complementing each other could lead to the construction of a device consisting of several sensors, taking the best behavior of each sensor or weighting their records according to the solar altitude.

An interesting assessment of the behavior of the sensors is the correlation of each one with the professional one (see Figure 13). As previously seen in Figure 9, there is a threshold in the behavior of the sensors at 70 klx. It was, therefore, decided to carry out a correlation study for values below and above 70 klx. It is observed that for values below 70 klx, the correlation of OPT3001 and TSL2591 is above 80%. This good correlation becomes poorer for values above 70 klx. The correlation drops below 40% for the TSL2591 and VML7700 sensors. The negative correlation of OPT3001 for this zone is striking, reflecting its erratic behavior. Sometimes it stops working; other times, it provides a value far from the true one. These correlations show the ability of low-cost sensors to follow the values recorded by the professional, despite their limitations for high values of illuminance.

## 4. Conclusions

The starting question was whether a traditional luxmeter could be swapped with one or several cheap-networked sensors capable of obtaining illuminance data in a distributed manner and leveraging this extra data to create better illuminance models. All tested sensors show a lack of capability in the upper range (above 75 klx), be it an inability to measure (OPT3001), clipped results (TSL2591), or values under the expected ones provided by the EKO luxmeter (VEML7700). A certain bias is also observed in the sensors’ response to different sun altitudes, where OPT3001 has a good performance at low altitudes, which worsens as the solar altitude increases, while the TSL2591 and VML7700 show the opposite behavior. Overall, the low-cost sensors analyzed show good precision. There is also a bias that could be corrected using a mathematical function. The use of these types of sensors could vastly improve the collection of illuminance data; not only will the costs be radically reduced, but also, the use of IoT tools will leverage the recording and treatment of the collected data. Both aspects will help in the modeling of the solar energy available for natural illumination inside buildings.

Despite the short length of the measurement campaign, the obtained results are considered to be meaningful and positive. The final objective of the project is the deployment of an IoT luxmeter ecosystem that can be seamlessly integrated with other data sources, either collocated or distributed. The first step is the development of a low-cost luxmeter capable of robustly offering similar results to a professional one. For this reason, before carrying out a complete development of the device from its most basic electronics, the accuracy of several existing solutions on the market has been analyzed in the hope of reusing them. Until a series of measurements spanning several years is obtained, the development of a mathematical model to calibrate the existing low-cost alternatives is ruled out.

The main future line of work will be the use of a neutral-density filter (ND) to reduce the illuminance range of sunlight. This type of filter, used in photography, reduces the intensity of all wavelengths equally. In other words, the total amount of light reaching the photosensor is reduced. In this way, photographs are obtained without burning caused by overexposure. Neutral density filters can be found that reduce the amount of light by up to 100,000 times (ND 100,000). In this case, the solar illuminance would range from 0 to 1 lx. Obviously, such an extreme filter would be poorly adapted to the response range of the sensor. In the case of selecting a filter that would let a quarter of the sunlight through (ND4), the illuminance received by the sensor would be between 0 and 25 klx. In this range, the sensors have a linear behavior—a good feature for calibrating the sensors using simple linear regression. Hence, sensors with a lower illuminance range that were discarded for this work could be used. In addition, by constraining the amount of incident illuminance, the solar energy received by the device would also be lower, and so would the heating.

Another future line is the use of several units of the same sensor. This will help ensure the repeatability of the measurements as well as the conclusions that can be drawn. In fact, this approach will help to diagnose whether the saturation problem of the OPT3001 is produced because of the manufacturer’s algorithms or because of a faulty unit. In this work, priority was given to testing a wider variety of sensors to check the feasibility of the idea. If negative results had been obtained, the continuation of this work would have been discouraged.

## Figures and Tables

**Figure 1 sensors-22-07107-f001:**
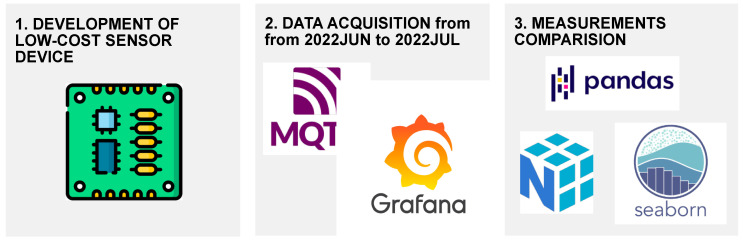
Workflow diagram.

**Figure 2 sensors-22-07107-f002:**
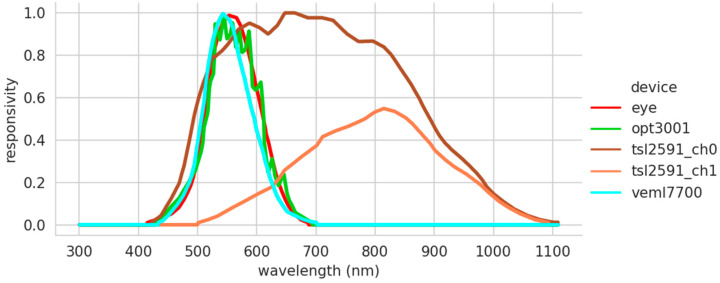
Spectral response of human eye and selected sensors.

**Figure 3 sensors-22-07107-f003:**
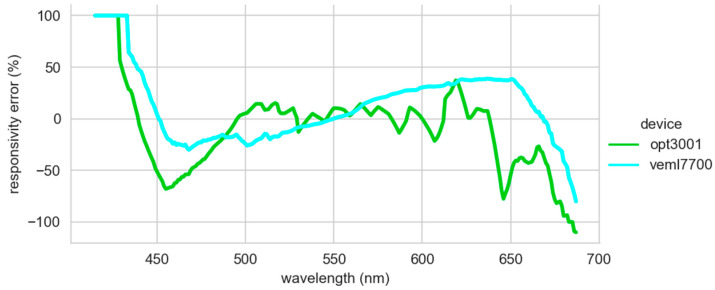
The relative error in the responsivity between low-cost sensors and the human eye.

**Figure 4 sensors-22-07107-f004:**
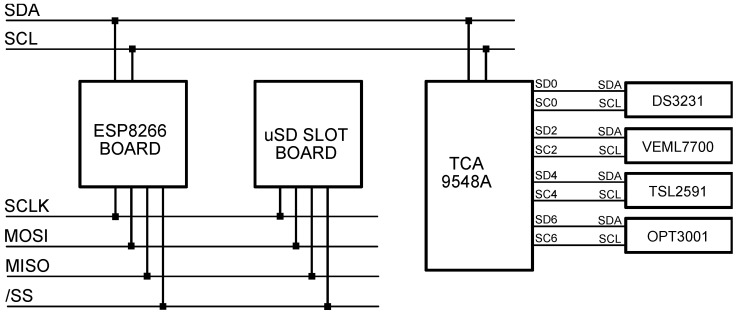
High-level circuit schematic of the PCB sensor board.

**Figure 5 sensors-22-07107-f005:**
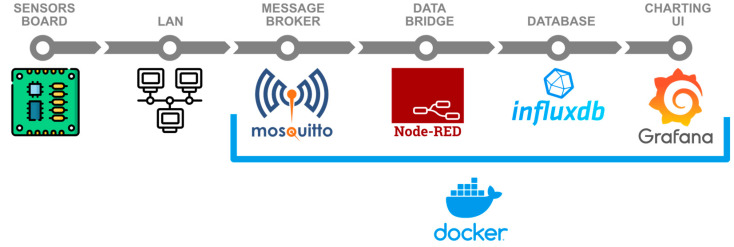
Block diagram of the end-to-end datapath.

**Figure 6 sensors-22-07107-f006:**
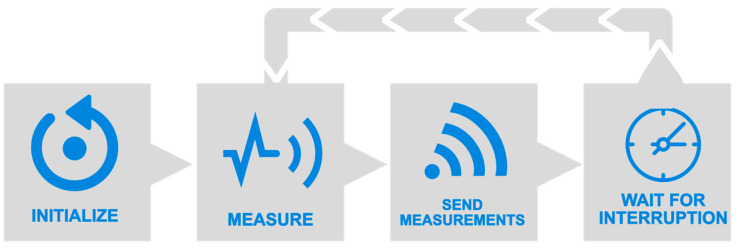
The high-level flowchart of the program running in the microcontroller.

**Figure 7 sensors-22-07107-f007:**
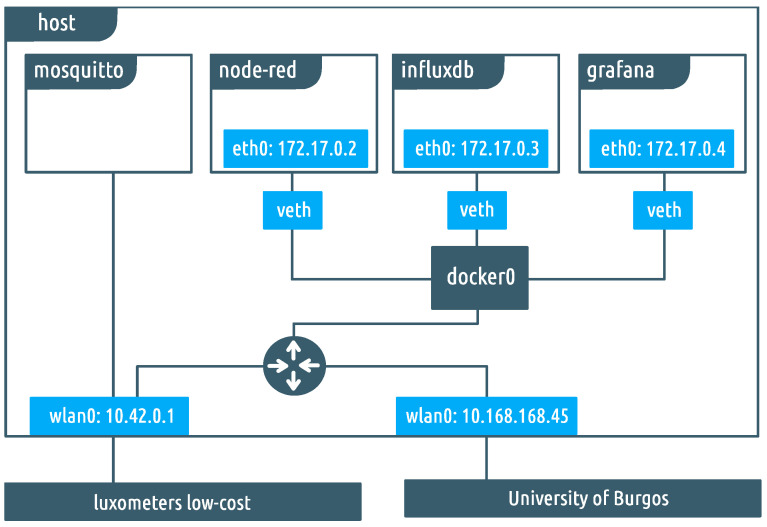
Detailed diagram of the containers’ infrastructure.

**Figure 8 sensors-22-07107-f008:**
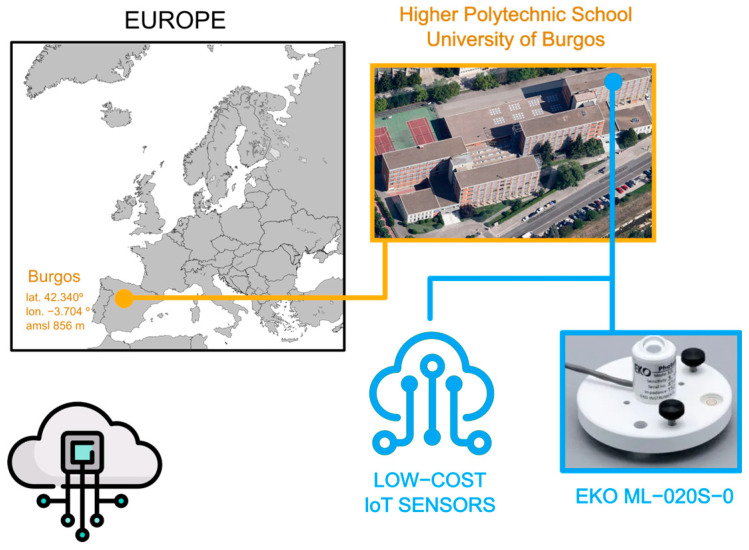
Experimental facility location at the High Polytechnic School of University of Burgos, Spain.

**Figure 9 sensors-22-07107-f009:**
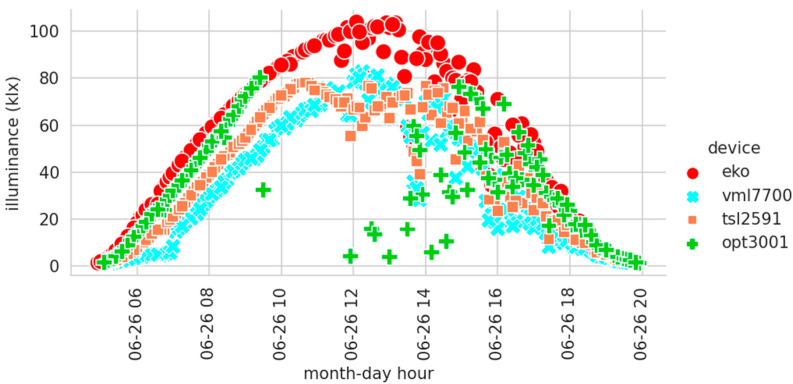
Illuminance measurements for 26 June 2022.

**Figure 10 sensors-22-07107-f010:**
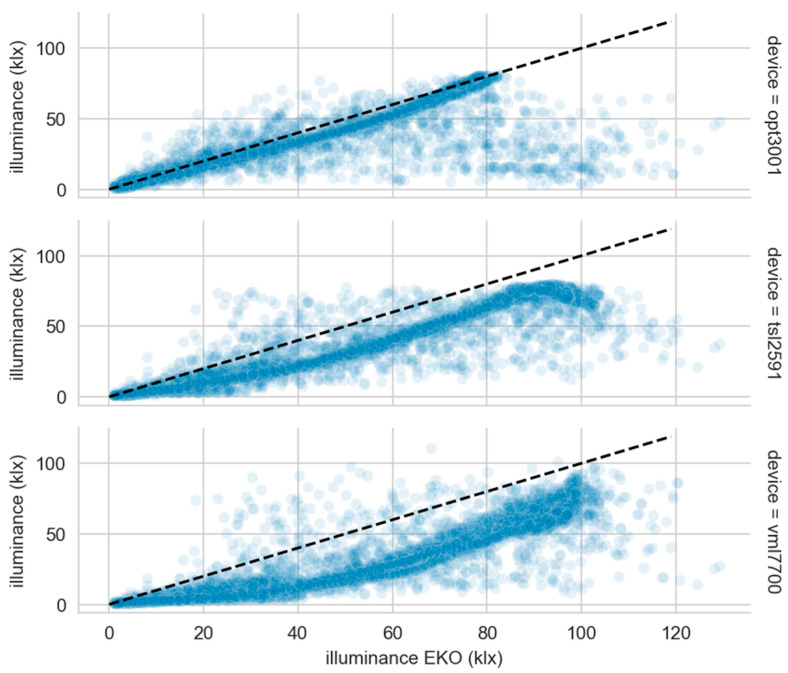
Illuminance low-cost sensor measurements versus professional one.

**Figure 11 sensors-22-07107-f011:**
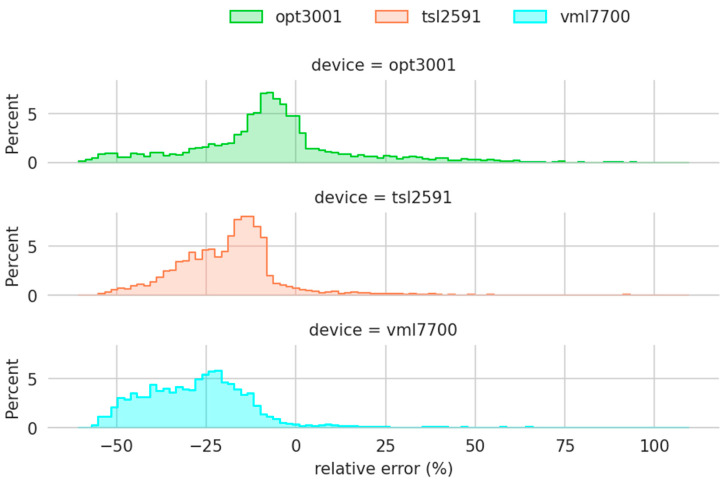
Histogram of the relative errors.

**Figure 12 sensors-22-07107-f012:**
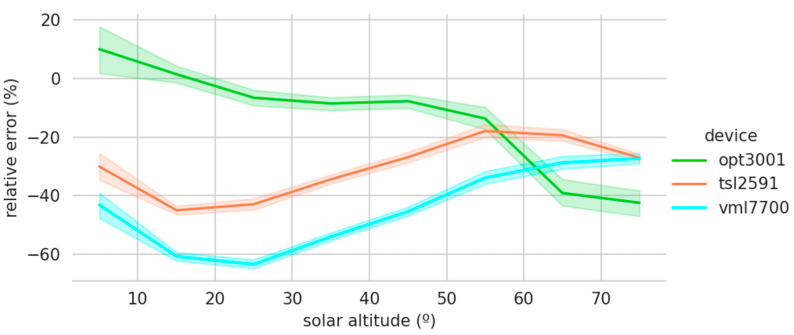
Relative errors grouped by solar altitude.

**Figure 13 sensors-22-07107-f013:**
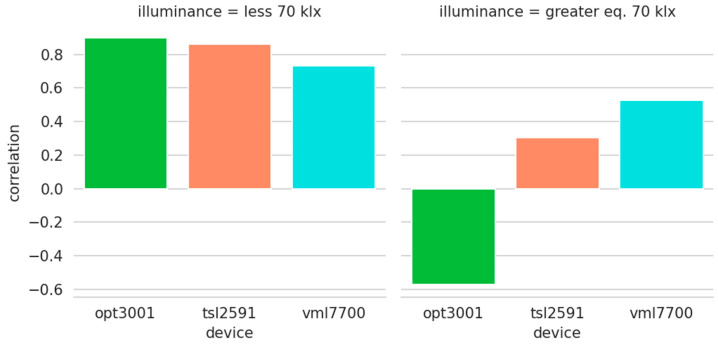
Correlation between low-cost sensors and professional one.

**Table 1 sensors-22-07107-t001:** Specifications of professional luxometer.

Sensor	Operating Voltage Range	Maximum Illuminance (klx)	Temperature Response −10 to 50 °C
ML-020S-O	0 to 30 mV	150	0.4%

**Table 2 sensors-22-07107-t002:** Low-cost illuminance sensor modules.

Sensor	Operating Voltage Range (V)	Operating Temp. Range (°C)	Maximum Illuminance (klx)	Light Spectrum
VEML7700	2.5 to 3.6	−25 to +85	120	visible
TSL2591	2.7 to 3.6	−30 to +70	88	visible + IR
OPT3001	1.6 to 3.6	−40 to +85	83	visible

## Data Availability

The data that support the findings of this study are available from the corresponding author, upon reasonable request.

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
