# Peer review of "A Low-Cost Luxometer Benchmark for Solar Illuminance Measurement System Based on the Internet of Things"

_sensors, 2022, doi:10.3390/s22197107_

Round 1
Reviewer 1 Report
This paper designed a net-worked wireless data acquisition device integrating the low-cost sensors, acquainted of some data during the summer months of 2022, and compare of the low-cost sensors measurements with the professional luxmeter. This work is interesting. The following questions should be considered.
1. Figure 2 (Spectral response of human eye and selected sensors) is questionable. Why responsivity is higher than 1.0?
2. Strictly, figure 4 is not a drawing of the PCB board, it is a schematic circuit. It is hard to understand the design in detail.
3. How to correct the bias by using a mathematical function?
4. Comparison of the low-cost sensor measurements against the calibrated one in Fig. 10 is obscure.
5. Abstract should be revised.
Author Response
Q1. Figure 2 (Spectral response of human eye and selected sensors) is questionable. Why responsivity is higher than 1.0?
Thank you for your feedback. The figure was obtained using a digitizer from the technical data sheet (https://cdn-shop.adafruit.com/datasheets/TSL25911_Datasheet_EN_v1.pdf page 8). It was an error that has been corrected in the new version of the figure.
Q2. Strictly, figure 4 is not a drawing of the PCB board, it is a schematic circuit. It is hard to understand the design in detail.
It is not indeed a PCB drawing but a high-level schematic of the circuit with the purpose of representing its main blocks and connections in the clearest possible manner (actually, only power connections are omitted)). We have made some changes in the figure hoping they will help to make it clearer. With future iterations we have the intention to release both schematic and gerber files as well as source code as open hardware so anyone can replicate our design.
Q3. How to correct the bias by using a mathematical function?
Let X be the illuminance registered by the low-cost sensors and Y the illuminance registered by the professional one. Correcting the bias is finding some function f such that Y = f(X). There are a lot of tools that one can use to do the job from the classical ones (regression) or the more exotic (regression trees, splines, neural networks,...). The authors have preference for the classical ones. However, the mathematical modeling is left until more data is collected.
Q4. Comparison of the low-cost sensor measurements against the calibrated one in Fig. 10 is obscure.
The comparison has been updated (line 348-389).
Q5. Abstract should be revised.
After careful revision, the authors cannot find any error or any way to improve the abstract.
Reviewer 2 Report
It is a well written manuscript and within the scope of the journal with potential interest for readers. The literature review is rather limited. Authors should refer to similar studies that focus on lighting and spectrum measurement tools that has IoT capabilities, such as:
- Botero-Valencia, J. S., & Valencia-Aguirre, J. (2021). Portable low-cost IoT hyperspectral acquisition device for indoor/outdoor applications. HardwareX, 10, e00216.
- Botero-Valencia, J. S., Valencia-Aguirre, J., Durmus, D., & Davis, W. (2019). Multi-channel low-cost light spectrum measurement using a multilayer perceptron. Energy and Buildings, 199, 579-587. -Kim, S., Jahandar, M., Jeong, J. H., & Lim, D. C. (2019). Recent progress in solar cell technology for low-light indoor applications. Curr. Altern. Energy, 3(1), 3-17. - Botero-Valencia, J. S., Valencia-Aguirre, J., & Durmus, D. (2021). A low-cost IoT multi-spectral acquisition device. HardwareX, 9, e00173. -Bruzzi, M., Cappelli, I., Fort, A., Pozzebon, A., Tani, M., & Vignoli, V. (2021, August). Polycrystalline silicon photovoltaic harvesting for indoor IoT systems under red-far red artificial light. In 2021 IEEE Sensors Applications Symposium (SAS) (pp. 1-6). IEEE. Statistics can also be improved. For example, correlation coefficients should be reported and normality/equal variance assumptions should be tested. Reference formatting is inconsistent. Authors should refer to MDPI guidelines.
Author Response
Q1. It is a well written manuscript and within the scope of the journal with potential interest for readers. The literature review is rather limited. Authors should refer to similar studies that focus on lighting and spectrum measurement tools that has IoT capabilities.
All the references have been added to the text in lines 74-81. Also, the authors want to thank the citations. The ones related with the multispectral acquisition look really interesting.
Q2. Statistics can also be improved. For example, correlation coefficients should be reported and normality/equal variance assumptions should be tested.
The statistic section has been augmented (lines 348-389). A correlation coefficient analysis and a histogram of the relative errors have been added. A deeper statistical analysis is planned to be done with more data in a near future.
Q3. Reference formatting is inconsistent. Authors should refer to MDPI guidelines.
The authors followed the "Free Format Submission" (https://www.mdpi.com/journal/sensors/instructions). As it is stated "Your references may be in any style, provided that you use the consistent formatting throughout.". The style used was the Chicago Style. Nevertheless, the style has been changed to the Multidiscplinary Digital Publishing Institute one.
Reviewer 3 Report
The article entitled “A Low-Cost Luxometer Benchmark for Solar Illuminance Measurement System Based on The Internet of Things”, compares several low-cost commercial outdoor sensors (VEML 7700, TSL2591 and OPT3001) with a professional one (ML-020S-O) and introduces the design of an IoT-based platform which is deployed as a centralized point of data collection and processing.
The subject of this research work is utterly interesting, its novelty and adding to knowledge is nevertheless suggested to be further justified for improving the value of this paper by adding some information on this issue in the introduction section and reporting some more methods for smart measuring solar illuminance.
The methodology which is followed in the proposed approach is clearly defined and the system is assessed adequately permitting other researchers to reproduce certain aspects. Additionally, the methodology analysis, as well as the assessment results are enriched with an efficient number of properly presented figures, tables and charts.
The conclusions of the research and their association with the experimental results are satisfactory defined. Nevertheless, the authors are advised to include a discussion wherein the results will be thoroughly interpreted in perspective of the working hypotheses, and the findings of the research as well as their implications will be discussed in in the broadest context possible.
Finally, the paper is well-structured in general and written in appropriate English language according to the standards of the Journal, however some minor spell-checking is required and the in-text referencing should be adjusted to the standards of the journal.
Author Response
Q1. The conclusions of the research and their association with the experimental results are satisfactory defined. Nevertheless, the authors are advised to include a discussion wherein the results will be thoroughly interpreted in perspective of the working hypotheses, and the findings of the research as well as their implications will be discussed in in the broadest context possible.
Thank you for your feedback. We have extended our conclusions section with the advised discussion (lines 394-402) and we hope that the comparative analysis of the obtained results against the starting hypothesis are now better presented and clearer exposed.
Q2. Finally, the paper is well-structured in general and written in appropriate English language according to the standards of the Journal, however some minor spell-checking is required and the in-text referencing should be adjusted to the standards of the journal.
The authors followed the "Free Format Submission" (https://www.mdpi.com/journal/sensors/instructions). As it is stated "Your references may be in any style, provided that you use the consistent formatting throughout.". The style used was the Chicago Style. Nevertheless, the style has been changed to the Multidisciplinary Digital Publishing Institute one.
Spell has been checked, correcting the found errors.
Reviewer 4 Report
Natural lighting is essential in home automation applications. It contributes to better visual health, energy savings, and lower CO2 emissions, among other benefits. As a result, it is critical to measure illuminance as accurately and cost-effectively as possible. This study contrasts several low-cost commercial sensors (VEML 7700, TSL2591, and OPT3001) with a professional sensor (ML-020S-O), all of which were installed outside. In addition, as a centralized point of data collection and processing, a platform based on Internet of Things technology was designed and deployed. The summer months were chosen for the comparison. This is the most difficult situation for low-cost sensors because they are designed for indoor use and their operating range is less than that of sunlight. Every minute, the solar illuminance was measured. The obtained bias is, as expected, dependent on the solar height. In the worst-case scenario, this can reach 60%, but it is usually less than 40%. The authors report useful results because they achieve high precision in the recordings. Because of this systematic deviation, it is amenable to mathematical correction. As a result, the incorporation of additional sensors and data that can aid in the overall improvement of the precision and accuracy of this low-cost system is left as a future line of improvement. All in all, it is a good paper and I recommend it to be accepted. The quality of Figures 4-8 can be improved. It is better to add a paper organisation at the end of Section 1.
Author Response
The authors would like to thank all the good comments received.
Q1. The quality of Figures 4-8 can be improved.
Figures 4-8 have been improved.
Q2. It is better to add a paper organisation at the end of Section 1.
A paragraph with the organisation of the paper has been added (lines 82-88).
Round 2
Reviewer 1 Report
Authors repond my main comments.